# Research on Impermeability of Underwater Non-Dispersible Concrete in Saline Soil

**DOI:** 10.3390/ma15227915

**Published:** 2022-11-09

**Authors:** Baolin Guo, Chao Wang, Xianghui Ma, Ruishuang Jiang, Baomin Wang, Jifei Yan, Hang Liu

**Affiliations:** 1School of Civil Engineering, Dalian University of Technology, Dalian 116000, China; 2Transportation Research Institute of Shandong Province, Jinan 250100, China; 3Shandong Hi-Speed Company Limited, Jinan 250014, China; 4Shandong Research and Development Center of Bridge-Tunnel Maintenance Technology and New Material, Jinan 250100, China

**Keywords:** salt soil, non-dispersible underwater concrete, resistance to chloride ion permeability, granulated blast-furnace slag

## Abstract

The permeability of different strength grades of submerged non-dispersible concrete with different granulated slag admixtures in a saline soil environment simulated by different erosion solutions was investigated. The variation patterns of the chloride ion diffusion coefficient and pore characteristics were tested using NEL and MIP. The microscopic morphology of the specimens in different erosion environments and with slag doping was observed using SEM. The results showed that the impermeability of concrete in sulfate and complex salt environments was significantly reduced. The resistance of concrete to chloride ion penetration increased with the increase in strength grade, and the Cl− diffusion coefficient of C35 was 5–30% lower than those of C30 and C25 underwater non-dispersible concrete at 360 d. Meanwhile, the admixture of granulated blast-furnace slag optimized the pore size distribution and improved the matrix compactness and permeability.

## 1. Introduction

Underwater non-dispersible concrete (NDC/NUC) refers to self-compacting concrete [1] that does not disperse or segregate under water owing to the addition of one or several water-soluble polymer compounds as flocculants to provide the concrete with good adhesion, anti-dispersibility and water scour resistance [2,3]. In the 1970s, Siebtechnik Tema (Mülheim, Ruhr, Germany) mixed a type of ether polymer compound into concrete and developed concrete with excellent water scour resistance [4]. Later, researchers designated the ether compound as a flocculant and widely used it in the production of concrete, resulting in the creation of underwater non-dispersible concrete. In the 1980s, Japan, European and American countries also started to research flocculants and underwater non-dispersible concrete [5], which have been extensively applied in engineering construction [6]. In 1987, the UWB-I flocculant was successfully developed in China. With the in-depth study of the flocculation system, the flocculant independently developed in China has continually achieved promising results in practical engineering applications.

Saline soil is a general term for salinized soil, alkali soil, and salt and alkali soil. At present, China’s saline soil is mainly concentrated in six western provinces and regions, namely, Gansu, Ningxia, Qinghai, Inner Mongolia, Xinjiang and Shaanxi, which cover an area of about 1 million square kilometers [7,8]. Saline soil contains ions in strong corrosive media, such as SO_4_^2−^, Cl^−^ and Mg^2+^ [9,10,11], which cause corrosion damage in concrete and then affect the service life of buildings [12,13]. Cl^−^ can penetrate reinforced concrete structures to destroy the passivation film on the surface of steel bars and accelerate their corrosion [14,15]. After SO_4_^2−^ enters the concrete, it absorbs water, crystallizes and expands, leading to concrete cracking and damage [16].

When a transmission line is constructed in a saline soil area, part of the cast-in-place concrete pile should be poured under water. The replacement of ordinary concrete with underwater non-dispersible concrete for underwater pouring can effectively improve the water scour resistance and anti-dispersibility of the cast-in-place pile [17]. However, there are also durability problems, such as chloride penetration resistance and water penetration resistance. Some researchers have pointed out that the pore distribution and pore size in concrete also affect the permeability of chloride ions [18,19]. Zhao Jing [20] et al. found that adding mineral admixtures (silica fume, etc.) can improve the fluidity of concrete mixtures [21,22] and improve the impermeability and durability of concrete [23,24]. Some researchers [25] found that introducing nanomaterials, such as a nano-admixture with CSH seeds, significantly improved the mechanical properties of concretes, especially at very early ages. In addition, slag powder is one of the most widely used slag admixtures in cement concrete by domestic and foreign scholars, and its mechanism and application in cement concrete have been comprehensively and thoroughly reported. However, its effect and mechanism in non-dispersed concrete are still unclear, especially when the environment is a complex salt solution.

This work aimed to study the impermeability of underwater non-dispersible concrete in a saline erosion environment and the modification effect of slag content on its impermeability by simulating a saline soil environment and using a microscopic testing method to explore its microscopic mechanism.

## 2. Materials and Methods

### 2.1. Materials

P·O 42.5R-type cement from Dalian Onoda Factory was used in this test. S105 granulated blast-furnace slag powder with an apparent density of 605 kg/m^3^ was used. The coarse aggregate was limestone with an apparent density of 2740 kg/m^3^ and particle size of 5~20 mm; the fine aggregate was natural river sand with an apparent density of 2.69 g/cm^3^ and fineness modulus of 2.80. The UWB-Ⅱ-type flocculant was selected with suspended turbidity < 150 mg/L and pH < 12; polycarboxylate superplasticizer was used with an effective solid content of 20% and water-reducing rate of 25%. The main chemical compositions of raw materials (mass fraction) are shown in Table 1. The main phase compositions of cement and slag are shown in Figure 1a,b, respectively.

### 2.2. Methods

#### 2.2.1. Preparation of Underwater Non-Dispersible Concrete

C25, C30 and C35 (corresponding water-to-cement ratios: 0.50, 0.45 and 0.40) underwater non-dispersible concrete specimens were prepared according to design rules for concrete mix ratios and test rules for underwater non-dispersible concrete. In particular, the standard referenced in the mix proportion design is the industry standard “Ordinary Concrete Mix Proportion Design Regulations” (JGJ55-2011), and the standard referenced in the process of concrete mixing and performance testing is the power industry standard “Underwater Non-dispersible Concrete Test Regulations” (DL/T5117-2000). In order to study the effect of strength grade on the impermeability of concrete, the underwater non-dispersed concrete had three strength grades, namely, C25, C30 and C35, corresponding to three water–binder ratios of 0.50, 0.45 and 0.40. Meanwhile, in order to study the effect of slag powder on the impermeability of underwater non-dispersed concrete, different dosages of slag powder were added to the concrete with the strength grade of C30. Mineral admixtures were mixed by the equal replacement method (SL1-20% dosage, SL2-40% dosage, SL3-60% dosage and SL4–80% dosage). The design and compressive strength for the concrete mix ratio are shown in Table 2. The specimens were cylinders of 100 mm in diameter and 100 mm in height.

Underwater molding method of the specimen: The test mold was put into a water tank containing fresh water or a salt solution, with the liquid level of the solution being 150 mm higher than the top of the test mold, and concrete was slowly poured in from the water surface to freely fall into the test mold. After pouring, the test mold was removed from the water tank and left to stand for 5 min at room temperature. After that, excess water was discharged by tapping on both sides of the test mold with a wooden hammer. After scraping the surface of the specimen with a spatula, the specimen was put back into water and cured for 48 h before form stripping. The concrete specimens were put back into fresh water (denoted by S), chloride solution (LY), sulfate solution (SY) or compound salt solution (FY) to be cured until the specified age (20~25 °C).

#### 2.2.2. Preparation of Salt Solution for Simulating Saline Soil Environment

The ion type and concentration of the salt solution were determined in accordance with the relevant technical building codes in the saline soil area (GB/T 50942-2014 and GB 50021-2001). The saline soil in China is widely distributed, and the ion concentration varies greatly among different regions, but the saline soil in most areas is slightly corrosive. According to the actual situation of saline soil in China, industrial NaCl and Na_2_SO_4_ were selected to prepare three erosion solutions: chloride (Cl^−^: 50,000 mg/L), sulfate (SO_4_^2−^: 20,000 mg/L) and their combination (Cl^−^ 50,000 mg/L + SO_4_^2−^ 20,000 mg/L). Selecting this concentration of etching solution can not only accelerate corrosion but also prevent the concentration from being too high and deviating from the actual situation. Meanwhile, the three erosive salt solutions were replaced every 15 days.

#### 2.2.3. Test methods

1.Chloride penetration resistance test

Specimens eroded to a certain age in salt solution were treated with salt using a NEL-VJH vacuum water retention machine, and the chloride diffusion coefficient was measured by a NEL-PDU chloride diffusion coefficient tester. The test block was removed from the solution 7 d before the test. A section with a height of 50 mm was sampled by obtaining a drilling core, and its two end faces were cut and rubbed. Three specimens in each group were tested, and the average value of all data within 15% of each specimen was taken as the measured value. Anti-chloride ion test pieces are presented in Figure 2.

2.Water impermeability test

Cone specimens with a top diameter, bottom diameter and height of 175 mm, 185 mm and 150 mm were prepared. The water permeability height method was selected, and the water impermeability of the specimens was tested by an HS-40S concrete penetrometer (Dalian Jinghua Construction Instrument Co., Ltd., Dalian, China). Water resistance test pieces are presented in Figure 3.

3.SEM and MIP tests

A QUANTA 450 scanning electron microscope (FEI Corporation, Hillsboro, OR, USA) and Micromeritics AutoPore Ⅳ9500 Series high-performance automatic mercury porosimeter (Mac Instruments, Sandusky, OH, USA) were used to analyze the microscopic morphology and pore characteristics of specimens in different erosion environments and with different slag contents.

## 3. Results and Discussion

### 3.1. Study on Cl^−^ Permeability Resistance of Concrete in Different Salt Environments

Figure 4 shows the Cl^−^ diffusion coefficient of the C30 strength-grade underwater non-dispersible concrete subjected to erosion by different salt solutions. The results show that the Cl^−^ diffusion coefficient of non-dispersible concrete gradually decreased with the prolongation of curing time. Compared with 28 days, the chlorine permeability resistance coefficient of C30 concrete in fresh water, sulfate, chloride salt and compound salt environments decreased by 38.27%, 30.82%, 36.84% and 28.28%, respectively, after curing for 360 days. At 28 d, the Cl^−^ diffusion coefficient of C30 concrete in different saline and freshwater environments was basically the same. From 90 d to 180 d, the Cl^−^ diffusion coefficient of concrete in sulfate and compound salt environments was higher than that in fresh water and 13.70% and 15.01% higher than that in the freshwater environment at 360 d, respectively. The Cl^−^ diffusion coefficient of concrete in the chloride salt environment was basically the same as that in fresh water.

SEM images of concrete at 360 d in fresh water and different salt solutions are shown in Figure 5. There is no evidence of ettringite or Ca(OH)_2_ production in the concrete in chloride solution (c). Compared with the non-dispersible concrete poured into fresh water (a), the microscopic characteristics are similar. Therefore, it can be seen that Cl^−^ had basically no effect on the Cl^−^ permeability resistance of concrete. In sulfate (b) and compound salt (d) environments, a large amount of acicular ettringite has been produced on the concrete surface, and the cement matrix is dilated and cracked, causing serious damage to the concrete structure. In chloride salts, the solubility of AFt is high, and Cl^−^ reacts with concrete hydration products such as Ca(OH)_2_ and C_3_A to generate hydrated chloraluminate, which delays the formation of AFt. The higher the concentration of Cl^−^, the more significant the effect [26]. SO_4_^2−^ and other ions in sulfate and compound salts enter the concrete, react with cement hydration products such as CH and C_3_A to generate dihydrate gypsum, ettringite, etc., and produce high crystallization stress, which promotes the generation and development of matrix cracks and then leads to the decline in the impermeability of concrete [27,28].

### 3.2. Influence of Strength Grade on Chloride ion Impermeability of Underwater Non-Dispersible Concrete

Figure 6a–c show the chloride ion diffusion coefficients of concrete with different strength grades subjected to sulfate, chloride salt and composite salt erosion, respectively. Through comparative analysis of the three figures, it can be seen that the Cl^−^ diffusion coefficient of concrete decreases with the increase in strength grade. As can be seen in Figure 6a, for the Cl^−^ diffusion coefficient at 28 d in the sulfate environment, the C30 value is 14.87% lower than that of C25 concrete, and the C35 value is 4.86% lower than that of C30 concrete. As can be seen in Figure 6b, for the Cl^−^ diffusion coefficient at 360 d in the chloride salt environment, the C30 value is 18.35% lower than that of C25 concrete, and the C35 value is 29.57% lower than that of C30 concrete. Similarly, for the Cl^−^ diffusion coefficient at 28 d in the composite salt solution, the C30 value is 18.49% lower than that of C25 concrete, and the C35 value is 1.02% lower than that of C30 concrete. It can be seen that the higher the strength grade, the lower the water-to-cement ratio, the higher the contents of cement and admixtures, the higher the amounts of hydration products, the denser the concrete structure, and the less easily erosive ions in the salt solution invade its interior.

### 3.3. Influence of Slag Powder Content on Chloride Permeability Resistance of Underwater Non-Dispersible Concrete

Figure 7a–c show the Cl^−^ diffusion coefficients of underwater non-dispersible concrete with different slag powder contents subjected to erosion by sulfate, chlorine salt and compound salt, respectively. As can be seen in Figure 7, the Cl^−^ diffusion coefficient of concrete decreases with the increase in slag powder content. The chloride diffusion coefficient of concrete with 60% slag powder content in the sulfate environment (a) is reduced by 6.49% compared with that without slag powder; at 360 d, the Cl^−^ diffusion coefficient with 80% slag powder content is reduced by 60.23% compared with that without slag powder. However, in the chloride salt environment (b) and compound salt environment (c), the Cl^−^ diffusion coefficient of concrete first decreases with the increase in slag powder content. When the slag powder content reaches 60%, the Cl^−^ diffusion coefficient of concrete is the minimum and increases to some extent when the slag powder content is 80%. This is because the cement content in concrete is reduced due to the high mineral powder content and insufficient strength development in the later stage. In the chloride salt environment, the Cl^−^ diffusion coefficient of concrete with 60% slag powder content is 49.29%, 67.15% and 68.27% lower than that of concrete without slag content at 28 d, 90 d, 180 d and 360 d, respectively. In the compound salt environment, the Cl^−^ diffusion coefficient of concrete with 60% slag powder content is 45.61%, 68.33%, 74.84% and 65.64% lower than that of concrete without slag content at 28 d, 90 d, 180 d and 360 d, respectively.

Figure 8a,b show the microscopic morphology of underwater non-dispersible concrete in the compound salt environment at 360 d without slag powder and with 60% slag powder content. It can be seen in the figure that acicular ettringite crystals are produced in the concrete without admixtures but not in the concrete with 60% slag powder content. Due to the secondary hydration reaction between cement hydration products CH and active SiO_2_ and Al_2_O_3_ in slag powder, calcium silicate hydrate gel is generated. The binding and adsorption effect of concrete on Cl^−^ is enhanced, the structure is denser, and its impermeability is improved [29].

### 3.4. Results Analysis of Water Impermeability Test for Underwater Non-Dispersible Concrete

Table 3 and Figure 9 show the results of the water penetration height of concrete with different strength grades in different solutions. As can be seen in the figure, the average water penetration height of C30 and C35 concrete in the composite salt decreased by 1.67% and 3.17%, respectively, at 28 d compared with the freshwater environment. After curing for 360 days, the average water penetration height of C30 and C35 concrete in the composite salt increased by 5.60% and 3.36%, respectively, compared with the freshwater environment. The results show that, at the early stage of curing, ions in the compound salt did not significantly affect the impermeability of concrete. At the later stage, SO_4_^2−^ and Cl^−^ invaded and reacted to generate AFt and hydrated calcium chloroaluminate, respectively [30], further eroding concrete. After curing for 360 days, the water penetration height of concrete in the salt solution was higher than that in the freshwater environment, which indicates that the composite salt environment restricted the improvement of the impermeability of NDC. In addition, compared with the corresponding curing for 28 days, the average water penetration height of concrete was reduced, among which the average water penetration height of concrete in the freshwater solution decreased by about 10% at 360 d, and that of concrete in the compound salt solution decreased by about 4%. It can be seen that the water impermeability of NDC is constantly developing.

### 3.5. Analysis of Pore Structure Characteristics of Underwater Non-Dispersible Concrete in Compound Salt Environment

Figure 10 shows the pore size distribution curve of concrete in fresh water, in composite salt solution and with 60% slag powder after curing for 180 days. The results show that the pore size distribution curve of concrete in the composite salt solution is similar to that in fresh water after curing for 180 days, but the pore size distribution is slightly better than that in fresh water. The porosity of concrete in the composite salt environment is 3.74% higher than that in fresh water. In the composite salt environment, ettringite is distributed in the micropores of concrete and the slurry aggregate interface area, and the formation of AFt in the early stage makes the concrete structure compact; however, due to the constant generation of ettringite, concrete expands and forms a large number of cracks, which further become a new channel for the diffusion of external chloride ions, increasing the porosity of concrete and the number of harmful holes and enhancing the permeability of concrete [30]. In the composite salt environment, the addition of slag powder decreases the pore size of underwater non-dispersible concrete, and the porosity decreases by 8.93% compared with that without slag powder. Due to the pozzolanic activity of slag powder, the hydration product CH in concrete is reduced, and the interface transition zone between aggregate and cement is optimized. The pore size is decreased, the porosity is reduced, the pore structure is obviously optimized, and the internal structure is more compact. C-S-H gel has a good adsorption effect on Cl^−^, and the aluminum phase in the slag powder can enhance the binding ability of Cl^−^ and cement slurry, reduce the penetration rate of Cl^−^ and improve the impermeability of concrete.

## 4. Conclusions

In this study, a sulfate solution, chlorine salt solution and composite solution of both were used to simulate a saline soil environment to study the microstructures of underwater non-dispersible concrete with different strength grades, in different erosion environments and at different erosion ages, as well as the change rules of chloride ion impermeability, water resistance and porosity. Additionally, this paper also explores the influence of different slag powder contents on the impermeability and resistance to chloride ions of underwater non-dispersible concrete, and the following conclusions are made:(1)In a freshwater environment and salt solution environment, with the increase in age, the chloride ion diffusion coefficient of underwater non-dispersible concrete gradually decreases, and the average water seepage height of concrete gradually decreases.(2)The chloride salt environment has no obvious influence on the impermeability of underwater non-dispersible concrete. The difference in the Cl^−^ diffusion coefficient compared to fresh water is less than 5%, and there is no obvious difference in morphology or pore size distribution. Sulfate reduces the impermeability of concrete. The Cl^−^ diffusion coefficient of concrete at 360 d in sulfate and composite salt solutions is 6%~15% higher than that in fresh water.(3)The improvement caused by the increased strength grade is conducive to the enhancement of the impermeability of underwater non-dispersible concrete. The Cl^−^ diffusion coefficient of C35 is 5%–30% lower than those of C30 and C25 underwater non-dispersible concrete at 360 d.(4)The addition of slag powder can significantly improve the impermeability of underwater non-dispersible concrete. The optimal dosage is 60%, and the Cl^−^ diffusion coefficient of concrete is the lowest, which is 45%~70% lower than that of the blank sample.

In this study, the test volume of the water penetration resistance test of underwater non-dispersed concrete was small, which led to a lack of systematic test data. Concrete with multiple strength grades can be tested to study the effect of strength grades on the water penetration resistance of concrete; different amounts of slag powder can also be added to concrete to study the effect of slag powder content on the water penetration resistance of concrete.

## Figures and Tables

**Figure 1 materials-15-07915-f001:**
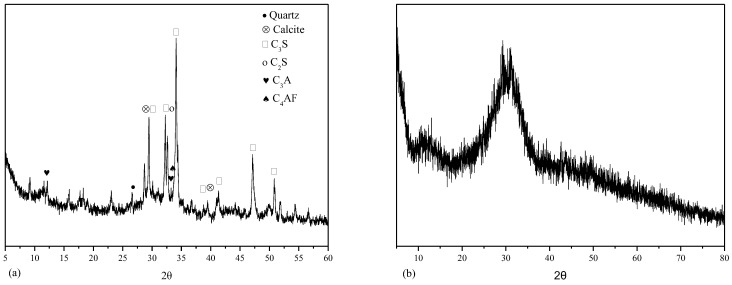
The phase composition of raw materials: (**a**) cement; (**b**) slag.

**Figure 2 materials-15-07915-f002:**
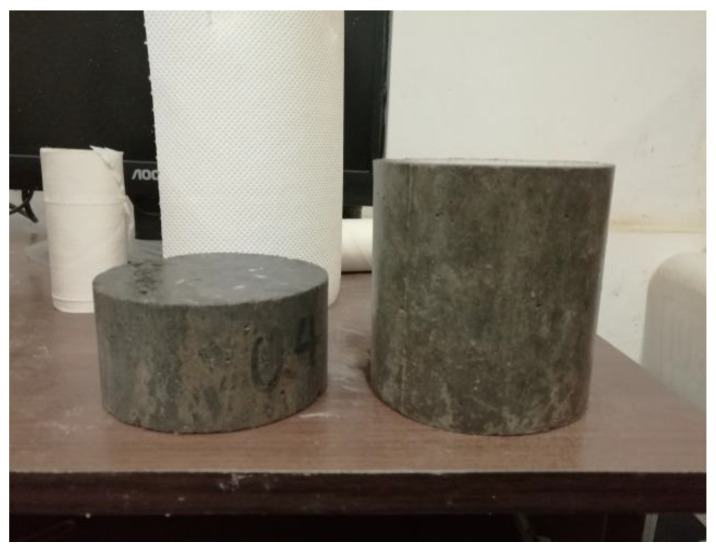
Anti-chloride ion test piece.

**Figure 3 materials-15-07915-f003:**
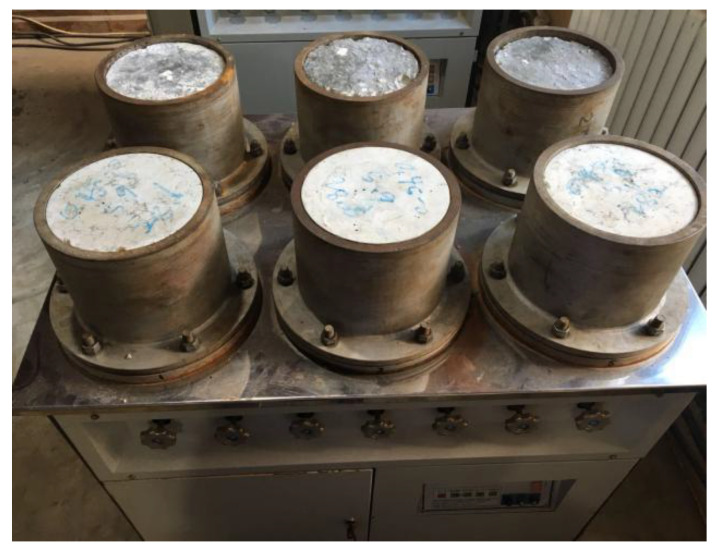
Water resistance test pieces.

**Figure 4 materials-15-07915-f004:**
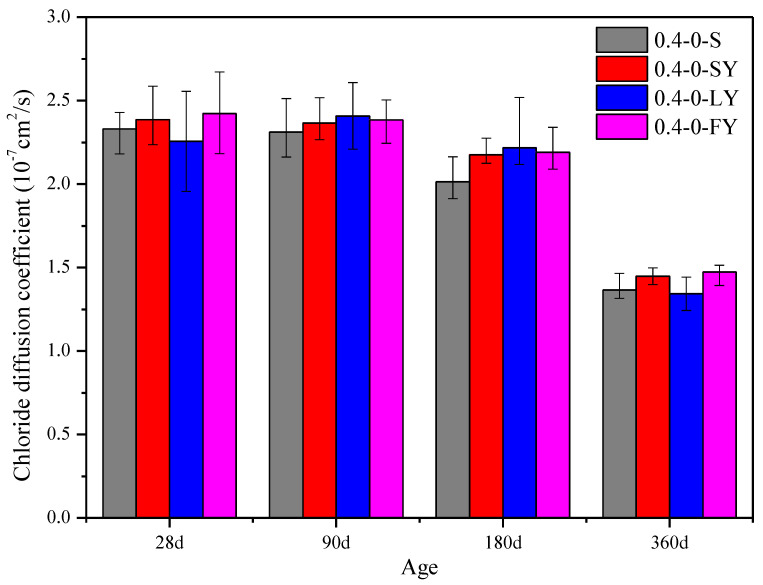
Chloride diffusion coefficient of C30 concrete.

**Figure 5 materials-15-07915-f005:**
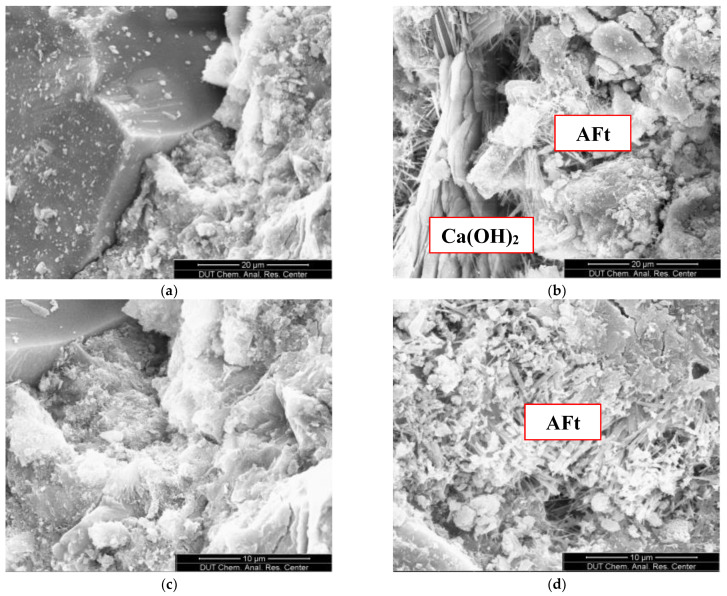
SEM images of concrete in different erosion environments: (**a**) fresh water, (**b**) sulfate, (**c**) chloride salt and (**d**) compound salt.

**Figure 6 materials-15-07915-f006:**
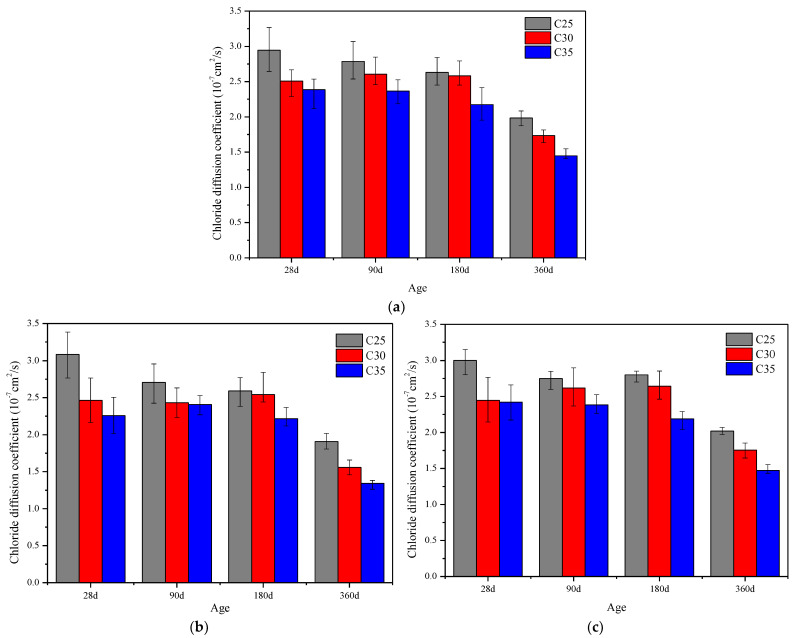
Diffusion coefficients of chloride ions in C25, C30 and C35 concrete in different aggressive environments: (**a**) sulfate, (**b**) chloride salt and (**c**) compound salt.

**Figure 7 materials-15-07915-f007:**
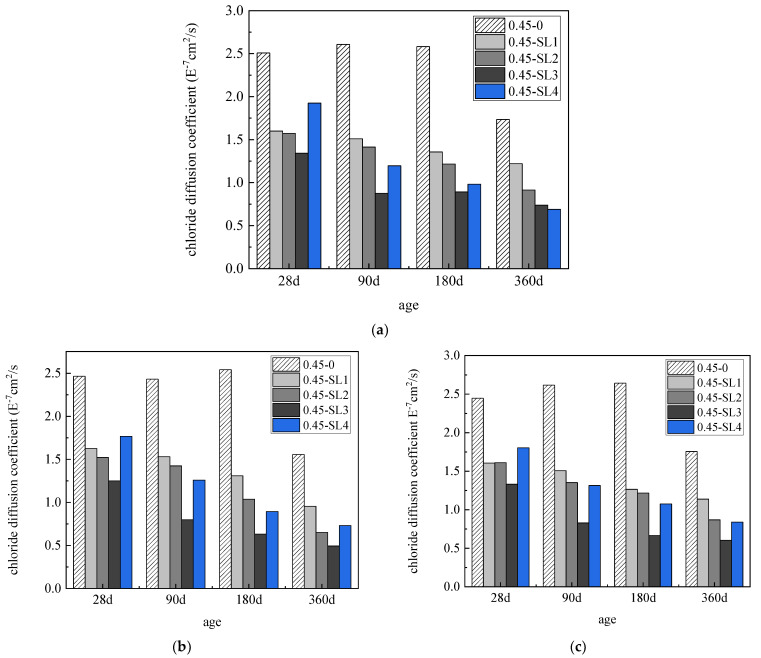
Chloride diffusion coefficients of concrete mixed with slag powder in different erosion environments: (**a**) sulfate, (**b**) chloride and (**c**) compound salt.

**Figure 8 materials-15-07915-f008:**
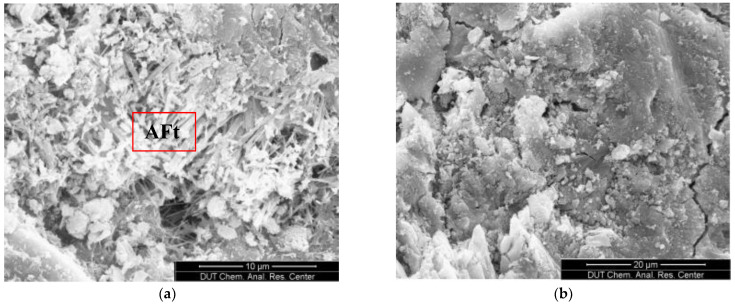
SEM images of concrete in a compound salt environment (**a**) without admixture and (**b**) with 60% slag.

**Figure 9 materials-15-07915-f009:**
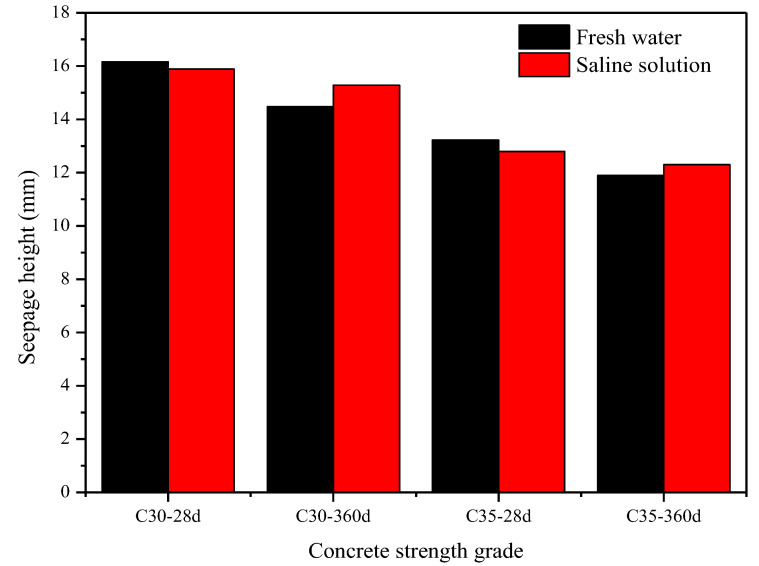
Concrete water penetration height test results.

**Figure 10 materials-15-07915-f010:**
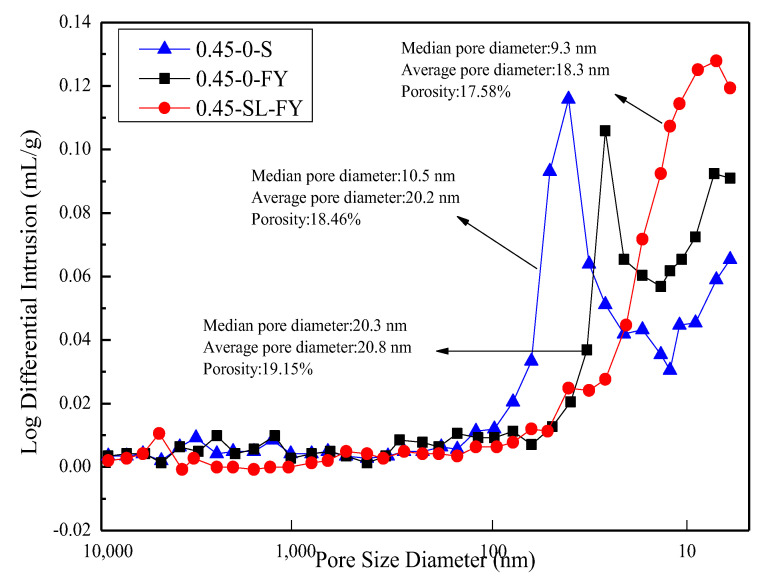
Log differential intrusion of concrete at 180 d.

**Table 1 materials-15-07915-t001:** Main chemical compositions of raw materials (mass fraction) (%).

Materials	CaO	SiO_2_	Al_2_O_3_	Fe_2_O_3_	SO_3_	MgO	Na_2_O	K_2_O	TiO_2_	MnO	Na_2_O_3_	Others
Cement	61.16	21.49	5.24	2.89	2.53	2.12	0.76	0.42	—	—	—	3.39
GGBS	40.08	30.31	14.99	0.39	3.22	8.60	—	0.50	0.84	0.40	0.32	0.35

**Table 2 materials-15-07915-t002:** Mixing ratio and compressive strength of concrete.

Strength Grade	No.	Water–Binder Ratio	Water (kg)	Binding Material (kg)	Cement (kg)	Slag Powder		Fine Aggregate (kg)	Coarse Aggregate (kg)	Water Reducer (%)	Flocculant (%)	Compression Strength (MPa)	
						Mass (kg)	Replacement rate (%)					28 d	90 d
C25	0.5-O	0.50	200	400	400	0	0	759	1006	1.0	3.5	35.9	40.2
C30	0.45-O	0.45	200	445	445	0	0	690	1035	0.5	2.8	41.6	47.9
	0.45-SL1	0.45	200	445	356	89	20	690	1035	0.5	2.8	47.6	52.0
	0.45-SL2	0.45	200	445	267	178	40	690	1035	0.5	2.8	50.4	51.0
	0.45-SL3	0.45	200	445	178	267	60	690	1035	0.5	2.8	47.1	50.3
	0.45-SL4	0.45	200	445	89	356	80	690	1035	0.5	2.8	44.6	48.2
C35	0.4-O	0.40	200	500	500	0	0	670	1005	0.5	2.5	44.3	50.6

**Table 3 materials-15-07915-t003:** Concrete water penetration height test results.

Strength Grade	Water–Binder Ratio	Curing Condition	Age	Seepage Height (mm)						
				A	B	C	D	E	F	Average Water Seepage Height
C35	0.40	Fresh water	28 d	17.40	13.08	8.45	12.59	13.26	14.57	13.22
			360 d	11.31	10.77	13.22	12.03	12.36	11.74	11.90
		Saline solution	28 d	12.96	13.41	13.92	11.54	10.76	14.23	12.80
			360 d	12.24	12.16	13.33	13.78	11.92	10.37	12.30
C30	0.45	Fresh water	28 d	14.81	21.12	14.39	14.58	17.11	14.96	16.16
			360 d	13.79	15.21	14.04	17.23	14.12	12.43	14.47
		Saline solution	28 d	17.49	16.83	12.42	13.57	16.80	18.27	15.89
			360 d	14.17	16.09	14.89	15.85	13.32	17.35	15.28

## Data Availability

The data presented in this study are available on request from the corresponding author.

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
