# Peer review of "Research on Impermeability of Underwater Non-Dispersible Concrete in Saline Soil"

_materials, 2022, doi:10.3390/ma15227915_

Round 1

Reviewer 1 Report

In the above-referred manuscript, the authors carried out an experimental study on the impermeability of underwater non-dispersible concrete in saline soil. Although new data and results are always of benefit and interest to the research community, this manuscript should be re-arranged or correct as follows:

1. line 88, “the specimen is a cylinder of 100mm in diameter and 100mm in height”, what is the standard the author followed? Please show and cite here.

2. The characters inside Figure 2 are so difficult to read. Please change the other colour.

3. Table 1, why is the sum of the chemical composition of both cement and GGBS not 100%? Please explain in the manuscript.

4. Figure 5, why does the author choose two different scales of SEM diagrams (10µm and 20µm) to compare? This is a little weird. Would the author show all SEM diagrams of specimens of 20%, 40% and 80% slag for more comparison and explanation?

Reviewer 2 Report

Here are my review comments:

GENERAL COMMENT

Although the manuscript is well written, the presentation of ideas is very cryptic.  It suffers from lack of specifics and details on several important items such as test procedures, design rules, test matrix, etc. I will highlight the specific instances in my specific comments.

SPECIFIC COMMENTS

1. Lines 49 to 60. This paragraph reviews the literature on research performed on chloride  penetration in concrete. Lines 61 to 67. This paragraph claims in general terms the current problems of chloride penetration without presenting any proof for these claims. The literature paragraph (Lines 49 to 60) does not support these claims.

2. Lines 61 to 67. In this paragraph, the authors propose to study the impermeability of concrete in saline environment. The authors do not clearly discuss the novelty of their research. It is imperative that the authors justify the research undertaken because it is novel and fills a research gap in the area of interest.

3. Lines 84 and 85. The authors are very cryptic about the design rules and test rules. What are these rules. Did they follow any standard or standards? These standards must be clearly stated.

4. Lines 86 and 87. Here the mineral admixtures dosages are identified. However these dosages were only used for mix C30 and not for mixes C25 and C35 (Please see Table 2). The authors do not give any explanation for this. 

5. Lines 103 and 104. What are the "relevant building technical codes"?

6. In section 2.2.2, the authors do not provide any justification for the choice of the concentration levels.

7. Nowhere in the manuscript the number of repetitions for any of the tests is discussed. The reliability of the data depends on the number of repetitions (number of identical specimens tested for each condition). Data presented in Figures 1, 3, 4, 8, and Tables 2 and 3 need to specify if the numbers are averages of how many repetitions. Statical analyses, e.g. ANOVA, of the data presented will enhance the confidence presented in the conclusions.

8. Section 3.5 - How was the pore size distribution measures/determined? What procedure was used?  

Reviewer 3 Report

In this manuscript, the impermeability of underwater non-dispersable concrete under salted erosion environment and the modification effect of slag content on its impermeability were investigated. In general, the manuscript was well written and some interesting and useful conclusions have been drawn. However, the article has some significant shortcomings that need to be corrected in its final version. Therefore, major revisions are needed in the following aspects:

(1) Abstract. Please add one or two major findings in the abstract.

(2) Introduction. At the end of the first section, the Authors indicated that “there are some problems in the research of chloride ion infilitration mechanism and permeability performance of underwater non-dispersable concrete". Please present the detailed in this regard.

 (3) Topic selection. There are a lot of types of materials and nano-materials which can be used in construction materials oi order to improve their properties or specific parameters, i.e. C-S-H nucleating agents. Why did the Authors select slag to modified the properties of cementitious composites ? The related explanation should be added. Moreover, one could also discuss and cite some new articles from this field, also published in Materials journal, for example:

Application of the CSH phase nucleating agents to improve the performance of sustainable concrete composites containing fly ash for use in the precast concrete industry, Materials 2021.

(4) Materials. Besides chemical composition plase provide phase composition of cement and GGBS used in the studies.

(5) Experiments. It is very important to include more details about the experimental setup. Therefore please provide photos showing specimens during mixing, molding, casting, testing, etc.

(6) SEM studies. Please provide magnification in SEM images from figs. 2 and 5. Moreover, the descriptions of the visible phases are missing in a few SEM images.

(7) Results. Charts with test results should contain error bars. Moreover, all charts should be made in colour, like in fig. 4.

(8) Summary and conclusions. In the summary of the paper perspectives for future research should be provided.

Round 2

Reviewer 2 Report

The authors have adequately responded to my comments.

Author Response

Thank you very much for your recognition.

Reviewer 3 Report

In my opinion, this article is highly recommended for the publication. However, not all amendments have been fully incorporated. Therefore additional minor improvements should be introduced:

(3) Topic selection. There are a lot of types of materials and nano-materials which can be used in construction materials oi order to improve their properties or specific parameters, i.e. C-S-H nucleating agents. Therefore, within the introduction and text, please add some recent references dealing new materials used to improve the parameters of cement composites. Some of this papers  could be found in MDPI database, for example:

- „Application of the CSH phase nucleating agents to improve the performance of sustainable concrete composites containing fly ash for use in the precast concrete industry, Materials 2021.

I strongly suggest to discuss and include in the references section this above paper.

(8) Summary and conclusions.

I would like to see more conclusion from such advanced studies.
